# Application of a Magnetic Platform in α6 Integrin-Positive iPSC-TM Purification

**DOI:** 10.3390/bioengineering10040410

**Published:** 2023-03-25

**Authors:** Pengchao Feng, Wenyan Wang, Wenhua Xu, Qilong Cao, Wei Zhu

**Affiliations:** 1Department of Pharmacology, School of Pharmacy, Qingdao University, Qingdao 266021, China; 2Institute of Regenerative Medicine and Laboratory Technology Innovation, Qingdao University, Qingdao 266021, China; 3Qingdao Haier Biotech Co., Ltd., Qingdao 266109, China; 4Advanced Innovation Center for Big Data-Based Precision Medicine, Beijing University of Aeronautics and Astronautics-Capital Medical University, Beijing 100083, China

**Keywords:** glaucoma, trabecular meshwork, regeneration, induced pluripotent stem cell, magnetic purification

## Abstract

The emergence of induced pluripotent stem cell (iPSC) technology has provided a new approach to regenerating decellularized trabecular meshwork (TM) in glaucoma. We have previously generated iPSC-derived TM (iPSC-TM) using a medium conditioned by TM cells and verified its function in tissue regeneration. Because of the heterogeneity of iPSCs and the isolated TM cells, iPSC-TM cells appear to be heterogeneous, which impedes our understanding of how the decellularized TM may be regenerated. Herein, we developed a protocol based on a magnetic-activated cell sorting (MACS) system or an immunopanning (IP) method for sorting integrin subunit alpha 6 (ITGA6)-positive iPSC-TM, an example of the iPSC-TM subpopulation. We first analyzed the purification efficiency of these two approaches by flow cytometry. In addition, we also determined cell viability by analyzing the morphologies of the purified cells. To conclude, the MACS-based purification could yield a higher ratio of ITGA6-positive iPSC-TM and maintain a relatively higher cell viability than the IP-based method, allowing for the preparation of any iPSC-TM subpopulation of interest and facilitating a better understanding of the regenerative mechanism of iPSC-based therapy.

## 1. Introduction

Glaucoma, the leading cause of irreversible blindness worldwide, is characterized by retinal ganglion cell loss and optic nerve damage [1,2]. In clinics, most patients with open-angle glaucoma (OAG) [3], the primary type of glaucoma, have significantly higher resistance to aqueous humor (AH) outflow than a healthy population [4]. As reported, 80–90% of this resistance originates from the disruption of the conventional outflow pathway comprising the trabecular meshwork (TM) and Schlemm’s canal (SC) [5]. Indeed, increasing evidence has demonstrated that a decline in TM cellularity is a critical pathogenic cause for OAG [6,7,8].

The emergence of stem cell technologies, especially the advancement in induced pluripotent stem cell (iPSC) technology, has provided new approaches to regenerating the decellularized TM of OAG. This type of autologous stem cell, generated through somatic reprogramming using Yamanaka factors, has been widely used in tissue regeneration [9] including TM regeneration. We and others have previously demonstrated that the TM-resembling cells derived from iPSCs display many TM features [10,11] such as the spindle-like cell shape, robust expression of TM biomarkers, dexamethasone-induced myocilin secretion, and dexamethasone-induced cross-linked actin network formation [12]. Transplantation of these iPSC-derived TM (iPSC-TM) cells can efficiently regenerate TM, restore AH outflow, and maintain intraocular pressure (IOP) homeostasis in several glaucoma models [13,14,15,16].

Two recent single-cell RNA sequencing (scRNA-Seq) studies have identified 12 distinct cell types in conventional outflow tissues [17,18] including myofibroblast- and fibroblast-like TM cells, pericytes, epithelium, endothelium, Schwann cells, and macrophages. We hypothesized that TM derived from differentiated iPSCs in the conditioned medium is a heterogeneous mixture. Thus, selecting the functional iPSC-TM subset from the heterogeneous mixture is of particular importance to developing an approach that can not only identify the specific iPSC-TM subpopulations, but also maintain the cell viability and longevity at a relatively high level.

Recent advances in cell separation methods and techniques such as immunomagnetic cell separation [19], fluorescence-activated cell sorting (FACS) [20], immunopanning (IP) separation [21], and density gradient centrifugation [22] have greatly facilitated the researchers’ understanding of the biological and molecular properties of cell subsets. Among these approaches, FACS is the most recognized high-yield and high-throughput approach through employing flow cytometry. However, the viability and longevity of the purified cells are low [23]. Immunomagnetic cell separation based on immunoreaction is an alternative approach, utilizes magnetic bead-conjugated antibodies (anti-cell surface proteins), and can make some cells have magnetism. Application of a high-gradient or open-gradient magnetic separator can purify these magnetic cells [24]. Moreover, the method allows the sorted cells to recover during passaging. The IP technique is an alternative immuno-approach that purifies cells using cell culture plates that have been pre-coated with some specific antibodies (anti-cell surface proteins). This approach could yield retinal ganglion cells at a higher viability than the magnetic sorting technique [21,25].

In this study, we employed two cell sorting approaches, immunomagnetic cell sorting and IP separation, and purified a subset of iPSC-TM cells that specifically express integrin subunit alpha 6 (ITGA6, an example of cell surface proteins). The results of the purification ratios and cell viabilities provided us with the optimized approach to obtain the ITGA6-positive iPSC-TM subset with high yield, high throughput, and high viability.

## 2. Materials and Methods

### 2.1. Human TM Cell Isolation and Culture

As previously described [10], human TM cells (hTM) were isolated from three human donors obtained from the Iowa Lions Eye Bank and Beijing Tongren Hospital. All clinical characteristics of the human donors including age, gender, race, ophthalmic information, and cause of death are provided in Table 1. The protocol for human tissue collection was approved by the Ethics Committee of Beijing Tongren Hospital and the Eye Bank Association of America following the tenets of the Declaration of Helsinki. 

Cells were cultured in Biopsy medium comprising MEMalpha (Gibco, New York, NY, USA), 10% fetal bovine serum (FBS; Gibco), and penicillin-streptomycin (Pen/Strep, Gibco) and kept in an incubator with a 5% CO_2_ atmosphere at 37 °C. After characterization according to the expression of TM biomarkers and the formation of a dexamethasone-induced, cross-linked actin network, cells in passages 5–8 were used for iPSC differentiation in this study.

### 2.2. Human iPSC Culture and Differentiation

Renal urethra epithelial cells were isolated from the urine of a human donor and reprogrammed using a non-integrating Sendai virus carrying four transcription factors, Oct4, Sox2, c-Myc, and Klf4 (Cyto TuneTM-iPS 2.0 Sendai Reprogramming Kit; Thermo Fisher, Franklin, MA, USA). iPSC colonies were successfully generated at days 9–28 of infection, transferred to plates pre-coated with 0.2% Matrigel (Corning, New York, NY, USA), and cultured in mTeSR-1 medium containing recombinant human basic fibroblast growth factor (rhbFGF) and human transforming growth factor beta (rhTGF-β; STEMCELL Technologies, Cambridge, MA, USA) for expansion. The matured human iPSC colonies were passaged after digestion using 5 mg/mL collagenase (Sigma-Aldrich, Saint Louis, MO, USA). For 20–30 generations, the iPSCs were Sendai virus-free. Written informed consent was obtained from the donor, and the renal cell reprogramming protocol was performed in accordance with the Chinese stem cell clinical research guidelines.

Medium conditioned by the hTM of donors 1–3 were collected, sterilized using a mixed cellulose ester membrane with 0.2 µm pores (Millipore, Bedford, MA, USA), and used for human iPSC differentiation [10,13,14,16]. After 25–30 days of differentiation, iPSC-derived TM cells, named iPSC-TM, were used to purify the ITGA6-positive/negative subpopulation.

### 2.3. Immunohistochemistry (IHC) Analysis

Cells growing on poly-D-lysine coated coverslips were fixed with 4% paraformaldehyde (PFA; Thermo Fisher) for 20 min. The fixed cells were rinsed in Dulbecco’s PBS (DPBS; 1×; 145 mM NaCl, 8.1 mM Na_2_HPO_4_·12H_2_O, 1.9 mM NaH_2_PO_4_·2H_2_O, pH 7.2–7.4; DPBS; Thermo Fisher) for 5 min, and incubated in the blocking solution (DPBS with 1% bovine serum albumin, BSA; Sigma-Aldrich) for 1 h. Cells were further incubated with the diluted FITC-labeled anti-human ITGA6 antibody (Miltenyi Biotec, Bergisch Gladbach, Germany; 130-097-245; 1:100) overnight. After rinsing with 1× DPBS, cell nuclei were stained with DAPI (Santa Cruz). The stained cells were mounted using Neutral Balsam (Solarbio, Beijing, China) and imaged by confocal microscopy (Nikon, Tokyo, Japan).

### 2.4. Magnetic Bead-Based Separation

Human iPSC-TM cells were rinsed twice with Dulbecco’s PBS (1×; 145 mM NaCl, 8.1 mM Na_2_HPO_4_·12H_2_O, and 1.9 mM NaH_2_PO_4_·2H_2_O, pH 7.2–7.4; DPBS; Gibco) and collected after digestion with 0.25% trypsin (Gibco) at 37 °C for 3 min. After centrifugation at 1000 rpm for 3 min, the cell pellet was resuspended with the sorting buffer comprising 1× PBS (Gibco) buffer, 0.5% bovine serum albumin (BSA; Sigma-Aldrich), and 2 mM ethylene diamine tetraacetic acid (EDTA; Sigma-Aldrich). After that, the cells were first incubated with a 20% (wt/vol) solution of FITC-labeled anti-human ITGA6 antibody (Miltenyi Biotec; 130-097-245) for 10 min at 4 °C and then with anti-FITC microbeads (Miltenyi Biotec, 130-048-701) for 15 min at 4 °C. The labeled cells on the microbeads were purified by passing through the LS column (Miltenyi Biotec, 130-048-401; void volume: 400 μL, reservoir volume: 8 mL) or LD column (Miltenyi Biotec, 130-048-901; void volume: 1.35 mL, reservoir volume: 7 mL), which was pre-placed on a magnetic separation rack (Miltenyi Biotec, 130-042-501) and pre-washed according to the manufacturer’s instructions, and named ITGA6-positive/negative iPSC-TM. In addition, cells without magnetism were collected after purification by passing through the LS column and designated as ITGA6-negative iPSC-TM.

### 2.5. Trypan Blue Staining Assay

The purified ITGA6-positive iPSC-TM cells were incubated with 0.4% (wt/vol) filtered trypan blue solution at room temperature for 3 min. Cell viability was analyzed using Countstar (Alit Biotech Co., Ltd., Shanghai, China).

### 2.6. Immunopanning (IP) Separation

Cell culture dishes (Corning) were pre-coated with Tris-HCL solution (50 mM, pH 9.5; Sigma-Aldrich) comprising the FITC-labeled anti-human ITGA6 antibody (Miltenyi Biotec, 130-097-245, dilution ratio: 1:500) at 4 °C overnight. Human iPSC-TM cells were washed with Dulbecco’s PBS (Gibco) and collected using 0.25% trypsin (Gibco). After centrifugation at 1000 rpm for 3 min, cells were resuspended in Biopsy media and seeded in the pre-coated cell culture dishes for 1 h at either 25 °C or 37 °C. The dishes were shaken every 5 min or 15 min. The supernatants were removed and centrifuged to collect ITGA6-negative iPSC-TM, and the adherent ITGA6-positive iPSC-TM cells were collected using 0.25% trypsin (Gibco).

### 2.7. Flow Cytometry Analysis

A total of 30,000–50,000 cells were rinsed with 1× Dulbecco’s PBS and suspended in 100 μL of 1× Dulbecco’s PBS (Gibco) containing 1% FBS (Gibco) and FITC-labeled anti-human ITGA6 antibody (Miltenyi Biotec, 130-097-245). After incubation for 30 min at 4 °C, cells were rinsed and resuspended with 500 μL 1× Dulbecco’s PBS (Gibco) containing 1% FBS (Gibco) for flow cytometry. The ratio of fluorescence-positive cells was analyzed by BD FACSCalibur (Becton Dickinson, Franklin Lakes, NJ, USA). Untreated cells were used as negative controls. The voltage and amplifier gain of FSC were set at E00 and 1.00, respectively. The voltage and amplifier gain of SSC were set to 340 and 1.00, respectively. The voltage of FL1 was 381.

### 2.8. Statistical Analysis

One-way ANOVA was performed for the statistical analysis of ITGA6-positive/negative ratios after positive/negative selection. Data were expressed as the mean ± SD. *p* values < 0.05 were considered to be statistically significant.

## 3. Results

### 3.1. Differentiation of hiPSCs into iPSC-TM

As previously reported [11], renal urethra epithelial cells were reprogrammed into iPSCs using Sendai virus-carrying Yamanaka factors (Oct4, Sox2, c-Myc, and Klf4). Meanwhile, hTM cells of three donors were isolated and cultured in vitro. The conditioned medium of these hTM cells was pooled, filtered, and used for iPSC differentiation (Figure 1A). As shown in Figure 1B, human iPSCs exhibited a typical embryonic stem cell-like morphology: a compact colony comprising highly packed cells that have a large nucleus-to-cytoplasm ratio. During differentiation, the iPSC morphology was changed into a TM-resembling cell shape. On day 3, the differentiated iPSCs began to expand, and the compact colony became loose. Cells gradually showed a spindle-like shape after 7 days of differentiation but were still smaller than the hTM cells. After 25 days of differentiation, cells kept growing until reaching a similar size as the hTM cells, which were designated as iPSC-TM cells and used for the following cell sorting.

Two recent scRNA-Seq have demonstrated the heterogeneity of hTM cells [17,18]. Our IHC staining results showed that only partial iPSC-TM expressed ITGA6 (Figure 1C, pointed by the arrows), indicating that iPSC-TM generated by the hTM-conditioned medium is also a heterogeneous mixture. To investigate the function of the different subpopulations of iPSC-TM, a proper approach to sorting these subpopulations is highly required.

### 3.2. A Magnetic Platform to Purify ITGA6-Positive iPSC-TM

We first applied ITGA6 antibody-conjugated magnetic beads to label ITGA6-positive cells and purified the labeled cells by a MACS magnetic platform. The procedure is shown in Figure 2A. After 25 days of differentiation, iPSC-TM cells in the red tube were labeled with the FITC-labeled anti-ITGA6 antibody and anti-FITC microbeads, transferred in the blue tube and referred to as pre-sorted cells. The non-magnetic cells were pulled down in the purple tube and named ITGA6-negative iPSC-TM cells. The ITGA6-positive iPSC-TM cells were further purified by passing through the LS column in a magnetic field. After removing the LS column, the purified ITGA6-positive iPSC-TM cells were collected in an orange tube and analyzed by flow cytometry (Figure 2B–F: 1st round sorting; Figure 2G–K: 2nd round sorting; and Figure 2L–P: 3rd round sorting).

In the histogram analyses of flow cytometry, the curves of the negative control, pre-sorted iPSC-TM cells, and post-sorted iPSC-TM cells were demonstrated using different colored tubes in the schematic illustration (Figure 2A). The FITC-positive threshold was determined according to the fluorescence intensity of unlabeled iPSC-TM cells. As shown in Figure 2B–P, the ratio of ITGA6-positive cells was significantly improved to 63.2% from 36.0% after the first round of purification and further improved to 72.3% and 87.7% after the second and third rounds of purification, respectively. However, after multiple rounds of purification, the morphology of the iPSC-TM cells was severely damaged (Figure 2K–P).

Finally, we quantified the purification efficiency of the magnetic platform based on 3–6 experimental repeats (Figure 2Q). Based on the FITC-positive threshold, the ratio of ITGA6-positive cells was about 0.42 ± 0.04 in the pre-sorted iPSC-TM samples, and further increased to 68 ± 2% (*p* < 0.01) after the first round of purification and to 86 ± 5% after multiple rounds of purification (*p* < 0.01 compared to the pre-sorted samples and *p* < 0.05 compared to the samples after the first round of purification).

### 3.3. The Magnetic Platform to Purify ITGA6-Negative iPSC-TM

As described earlier (Figure 2A), the ITGA6-negative cells in the purple tube can be easily collected after LS purification since these cells are non-magnetic. To further improve the ratio of ITGA6-negative cells, we applied the LD column to further exclude ITGA6-positive iPSC-TM cells by passing through a magnetic field (Figure 3A). In a representative experiment, the ratio of ITGA6-negative cells was significantly improved to 78.9% (Figure 3B–D) from 64.0% (Figure 2B,C). The morphological observation indicated that one round of LD purification did not lead to noticeable damage to the iPSC-TM cells (Figure 3E). In addition, we also quantified the data of six experimental repeats and observed that the ratio of ITGA6-negative cells could be significantly improved to 73 ± 2% from 61 ± 4% (*p* < 0.05; Figure 3F).

### 3.4. IP Method to Purify ITGA6-Positive/Negative iPSC-TM Cells

As reported, the IP method is feasible to purify many types of cells from a heterogeneous mixture [26]. Here, we also applied IP to purify the ITGA6-positive cells from the heterogeneous iPSC-TM cells (Figure 4A). Four conditions were employed to exclude the floating ITGA6-negative iPSC-TM cells: oscillation every 15 min at 37 °C (Figure 4B–E), oscillation every 5 min 37 °C (Figure 4F–I), oscillation every 15 min at 25 °C (Figure 4J–M), and oscillation every 5 min at 25 °C (Figure 4N–Q). However, the IP-based purification under these four conditions showed no significant effect on isolating the ITGA6-positive iPSC-TM cells (Figure 4T: pre- vs. post- at four different conditions: 39.2% vs. 38.4%, 74.0% vs. 64.3%, 61.1% vs. 47.2%, and 61.1% vs. 44.7%, respectively). In contrast, the ratio of ITGA6-positive cells appeared to be smaller after purification. In addition, oscillation severely damaged the iPSC-TM cells (Figure 4R–S). Meanwhile, we collected the floating cells to analyze the ratio of ITGA6-negative cells through flow cytometry. In comparison with the ratios of the ITGA6-negative cells before IP (Figure 4C,G,K,O), the ratio of ITGA6-negative cells was higher after IP-based purification (Figure 5E: pre- vs. post- under the four conditions: 60.8% vs. 94.05%, 26.0% vs. 57.7%, 38.9% vs. 50.4%, and 38.9% vs. 48.4%, respectively).

**Figure 3 bioengineering-10-00410-f003:**
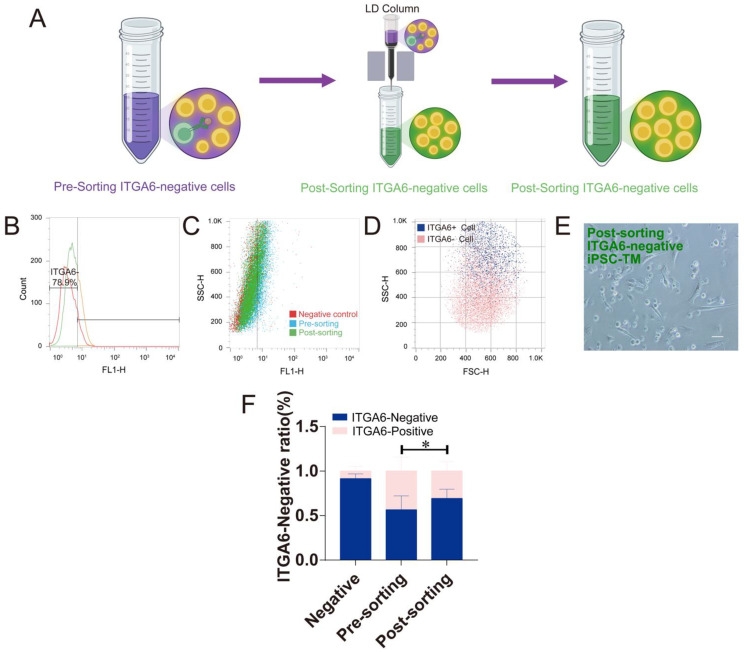
The magnetic platform to purify ITGA6-negative iPSC-TM. (**A**) Schematic illustration of the ITGA6-negative iPSC-TM purification process by the magnetic platform. The ITGA6-negative iPSC-TM after LS-based purification was collected in a purple tube and used for the LD-based purification. Post-sorted ITGA6-negative iPSC-TM in a green tube was used for the flow cytometry analysis. (**B**–**D**) The histogram analysis (**B**), fluorescence density plot (FL1-H vs. SSC-H; **C**), and FSC-H vs. SSC-H plot (**D**) showing the ratios of the ITGA6-negative cells in the negative control (red), pre-sorted iPSC-TM (blue), and post-sorted iPSC-TM (green). (**E**) A representative image showing the morphology of ITGA6-negative iPSC-TM after the LD-based sorting. Scale bars, 50 µm. (**F**) Quantification analysis of the ratios of ITGA6-positive cells (pink) and ITGA6-negative cells (blue) based on six experimental repeats. One round of LD-based purification significantly increased the ratio of ITGA6-negative cells (pre-sorting vs. post-sorting: 0.61 ± 0.04 vs. 0.73 ± 0.02, * *p* < 0.05). *p* value was analyzed by one-way ANOVA.

## 4. Discussion

Since the discovery of iPSC technology in 2006, human iPSCs have revolutionized the studies of human disease remodeling and regenerative medicine technologies, providing us with more opportunities for new drug discovery. New evidence from iPSC studies indicates the heterogeneity of iPSCs concerning their self-renewal capacity and pluripotency [27]. Moreover, the somatic cells derived from iPSCs such as cardiomyocytes, airway epithelial cells, and retinal ganglion cells have also been proven as a heterogeneous mixture [28]. Beyond this, our differentiation approach using the hTM-conditioned medium, a heterogeneous mixture comprising many types of cells, can generate TM-resembling cells from iPSCs, mixing not only with the undifferentiated cells, but also some other types of cells. The heterogeneous properties of iPSCs impede our understanding of how diseases are developed and may be treated. Therefore, new technologies for purifying different iPSC-derived subpopulations from a heterogeneous mixture are highly required in the accurate medicine field. 

MACS-based purification has also been widely used in sorting many cell types due to its simplicity, cheapness, and convenience. We have previously demonstrated a viable immunomagnetic sorting approach to depleting the undifferentiated cells from iPSC-TM based on the robust expression of SSEA-1 by stem cells using LD columns [13,14,16]. Compared to LD columns, the LS columns can be used for the positive selection of cells that strongly express the magnetically labeled surface antigen. We employed this technique (Figure 2, positive selection using the LS columns; Figure 3, negative selection using the LD columns) to purify the ITGA6-positive/negative iPSC-TM cells. We also evaluated this approach by analyzing the purification efficiency and cell viability of the purified cells. The results in Figure 2 and Figure 3 confirm that LS/LD-based magnetic purification after only one round enabled us to significantly increase the ratios of the ITGA6-positive/negative iPSC-TM cells. In comparison to multiple rounds of purification, the cells were still in a relatively healthy condition after one round of purification such as pre-sorted iPSC-TM (Figure 2; cell viability: 91.7% by the Trypan Blue Staining Assay). These observations demonstrated that MACS-based purification is feasible for sorting iPSC-TM expressing ITGA6 and ITGA6-negative cells. Indeed, applying the MACS system in sorting fibroblasts strengthened our conclusion. Like TM cells, fibroblasts are flat and spindle-shaped cells and play a supporting role in many tissues such as vessels and muscles [29,30]. Consistent with our findings, fibroblasts in the ductus arteriosus could be successfully isolated through a positive purification using the MACS system based on their robust expression of CD90 [29]. In addition, fibroblasts could also be negatively purified from the nerve tissue using the same system based on p75^NGFR^ expression in Schwann cells [30].

Based on these findings, this MACS-based purification can be applied as follows.

i.In the early 1950s, the TM was discovered to be an elaborate and complex tissue anatomically [31]. In short, it comprises three distinct layers with different structures and functions in aqueous humor drainage, indicating the heterogeneity of the TM. Recently, two new studies using scRNA-Seq have verified this heterogeneity, demonstrating 12 types of cells in the conventional outflow tissue [17,18]. Although researchers have investigated the different roles of cells in juxtacanalicular connective tissue or uveal meshwork [32,33], the functions of many other cell types such as Schwann cells, melanocytes, and T cells are still largely unknown. Our data using the MACS-based approach to purify ITGA6-positive iPSC-TM suggested that this purification method may be feasible to isolate the above subpopulations using different cell surface markers and investigate the roles of different cell clusters in controlling AH outflow and regulating IOP homeostasis.ii.Loss of TM cellularity, aberrant extracellular matrix remodeling, changes in the biomechanical properties of the TM, and mutations have been reported as risk factors for glaucoma [34,35]. For example, the first pathogenic mutation for primary open-angle glaucoma has been identified in the *myocilin* gene [36]. The aggregation of mutant myocilin can lead to a severe decline in TM cellularity due to endoplasmic reticulum stress [37,38]. To this end, it is very important to investigate how damage occurs in different TM subpopulations. Our MACS-based purification provides a simple method to generate TM subpopulations of glaucoma, which may facilitate us in determining the dysfunctional TM subpopulation at the earliest stage. The study can benefit not only the diagnoses but also the treatments for glaucoma patients in the early stages.iii.In recent times, some new glaucoma drugs have been identified that function primarily by modulating the TM cytoskeleton and the contractile tone of TM cells, their volume, and extracellular matrix deposition such as Rho kinase inhibitors [39], nitric oxide (NO) signaling regulators [40], latrunculins [41], and ion channel regulators [42]. Aside from pharmacologic treatments, gene therapy also holds a great promise in rescuing TM dysfunction [43,44,45,46]. However, which subpopulations of the TM that could be efficiently regulated by these new treatments are still elusive. The other side of answering this question would benefit the discovery of the proper delivery approaches for these new drugs/gene therapies. To this end, our MACS-based approach is feasible to address this question.iv.Moreover, we applied iPSC-TM in regenerating the damaged TM of several glaucoma models including Tg-MYOC^Y437H^ mice [13,14], GCα1^-/-^ mice, and aged human eyes [15]. As previously investigated, a common phenomenon of iPSC-TM after cell transplantation is that endogenous TM cells could be stimulated to proliferate. Aside from iPSCs, mesenchymal stem cells (MSCs) are also used in TM regeneration [47,48]. MSCs exist in the TM and are identified by analyzing the expressions of stem cell biomarkers [49]. In glaucoma animal models, the transplanted MSCs exhibit positive therapeutic effects on TM regeneration [47,50,51] including migration into the TM, secretion factors to recruit nesting-positive progenitors, and the stimulation of the cell proliferation of endogenous TM cells. Although encouraging, it is still elusive as to how the transplanted cells stimulate endogenous cell proliferation. Thus, our MACS-based purification could efficiently isolate different iPSC-TM or MSC subpopulations that are of interest and facilitate a better understanding of the mechanism of stem cell-based therapy.

Although promising, MACS-based purification still has several limitations. One is that after one round of purification, the ratio of ITGA6-positive cells varied from 63% to 71% (Figure 2Q), making it difficult to investigate the function of the iPSC-TM subpopulation. The primary reason is that the initial ratio of the ITGA6-positive cells in pre-sorted iPSC-TM varied between 36% and 50% (Figure 2Q), mainly due to our differentiation approach using the hTM of different viabilities. In the future, a new differentiation approach that could induce iPSC differentiation efficiently and generate a stable ITGA6-positive ratio should be developed. Another possibility is the non-specific labeling of an anti-ITGA6 antibody to some ITGA6-negative cells. Thus, the washing procedure should be strictly controlled. Moreover, cell damage after multiple rounds of purification (Figure 2) is attributed to the mechanical cue from the plunger during the LS-based purification. Although cells after one round of purification are healthy, it is still worth reducing the mechanical cue during purification. The third limitation of these antibody purification methods is receptor-mediated antibody internalization [52], which would impair their effectiveness in cell purification. Our approach should be optimized based on the biological half-life of α6 integrin. Alternatively, anti-β1 integrin antibodies could be involved to stabilize the heterodimer on the cell surface [53]. Another factor that may influence the effectiveness of these methods was the utilization of trypsin when we collected cells. As reported, trypsin can lead to the removal of the N-terminal domain of α6 integrin, and thus may affect the antibody binding sites [54]. Enzyme-free release of iPSC-TM should be considered.

Furthermore, an alternative approach, IP-based cell purification, was applied in this study. IP-based cell purification, designated to increase the ratio of ITGA6-positive cells, appears to trigger the opposite effect, which reduces the ratio of ITGA6-positive cells (Figure 4). One possibility for this phenomenon is the improper dilution ratio of the ITGA6 antibody. Until now, only one dilution ratio (1:500) was employed in our study. Due to the mild expression of ITGA6 in iPSC-TM, this dilution ratio should be increased. Another concern is that the ITGA6-negative cells could also adhere to the ITGA6 antibody-coated dishes more easily than the ITGA6-positive cells. If our speculation is correct, the supernatant should contain more ITGA6-positive cells than ITGA6-negative cells. However, the truth is that a decreased ratio of ITGA6-positive cells was found in the supernatant after purification (Figure 5). Indeed, our morphological observation that oscillation led to severe damage to ITGA6-positive iPSC-TM cells explained our quantification results. IP-based purification leads to a severe loss of ITGA6-positive cells, which is improper for sorting iPSC-TM subpopulations. However, it is undeniable that IP-based purification is feasible for sorting many other cell types such as retinal ganglion cells, neurons, and glia [21,26].

Compared to IP-based purification, we successfully applied a MACS system to purify the ITGA6-positive subpopulation in iPSC-TM and yielded ITGA6-positive cells with high viability. Likewise, utilizing different surface proteins allowed us to prepare any iPSC-TM subpopulation of interest. This study could not only facilitate our understanding of how diseases may be treated, but also benefits the development of accurate medicine.

## Figures and Tables

**Figure 1 bioengineering-10-00410-f001:**
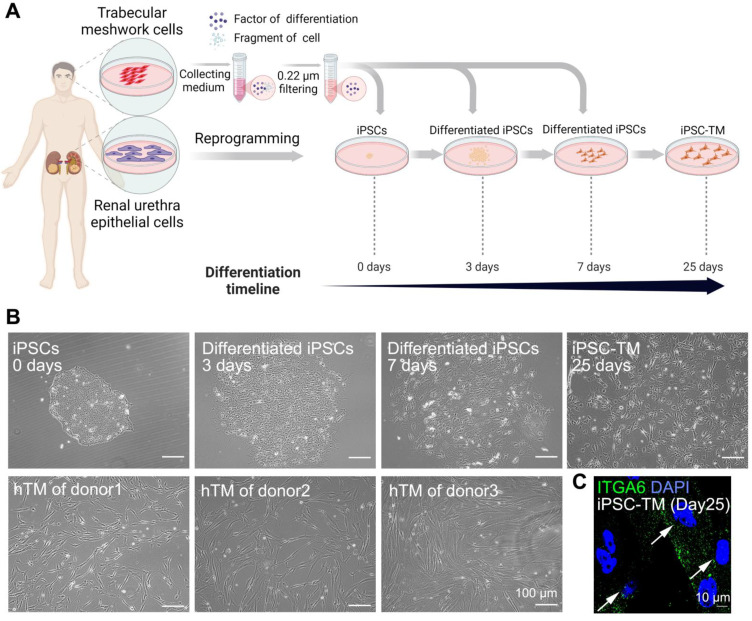
Differentiation of iPSCs into TM-resembling cells by the conditioned medium. (**A**) Schematic illustration of iPSC generation and differentiation by the conditioned medium approach. (**B**) Representative morphology of iPSCs, differentiated iPSCs for 3, 7, and 25 days, and hTM cells of three donors. Scale bars, 100 µm. (**C**) IHC staining of ITGA6 (green) and nuclei (blue) shows only partial iPSC-TM expressing ITGA6 (pointed by the arrows). Scale bars, 10 µm.

**Figure 2 bioengineering-10-00410-f002:**
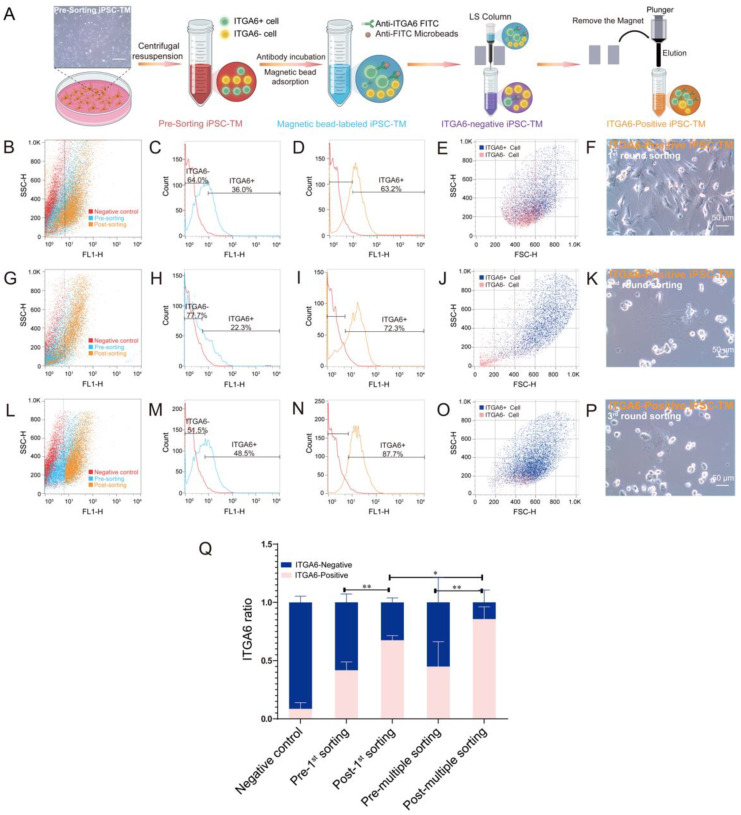
The magnetic platform to purify ITGA6-positive iPSC-TM. (**A**) Schematic illustration of the ITGA6-positive iPSC-TM purification process. The unlabeled iPSC-TM cells in the red tube were the negative control. iPSC-TM cells after labeling with FITC-conjugated anti-ITGA6 antibodies and anti-FITC microbeads were collected in a blue tube. ITGA6-negative iPSC-TM after the LS-based purification was collected in a purple tube, while ITGA6-positive iPSC-TM after the LS-based purification was collected in an orange tube. (**B**) A fluorescence density plot (FL1-H vs. SSC-H) depicting the distribution of the negative control cells (red), pre-first sorting cells (blue), and post-first sorting cells (orange). Each dot indicates an individual cell. (**C**,**D**) Histogram analyses of the negative control cells (red), pre-first sorting (blue) cells, and post-first sorting (orange) cells. The ratios of ITGA6-positive cells and ITGA6-negative cells are labeled in each plot. (**E**) A plot (FSC-H vs. SSC-H) indicating the distributions of the ITGA6-positive cells (blue) and ITGA6-negative cells (pink) in the post-first sorting samples. Each dot represents an individual cell. (**F**) A representative image showing the morphology of the ITGA6-positive iPSC-TM after one round of LS-based sorting. (**G**–**J**) Similar to panels (**B**–**E**) showing the flow cytometry results after two rounds of LS-based sorting. (**K**) An image showing the morphology of ITGA6-positive iPSC-TM after the second round of cell sorting. (**L**–**O**) Similar to panels (**B**–**E**) showing the flow cytometry results after three rounds of LS-based sorting. (**P**) An image showing the morphology of ITGA6-positive iPSC-TM after the third round of cell sorting. Scale bars, 50 µm. (**Q**) Quantification analysis of the ratios of ITGA6-positive cells (pink) and ITGA6-negative cells (blue) based on 3–6 experimental repeats. LS-based purification significantly increased the ratios of the ITGA6-positive cells (pre-first sorting vs. post-first sorting: 0.42 ± 0.042 vs. 0.68 ± 0.02, ** *p* < 0.01; pre-multiple sorting vs. post-multiple sorting: 0.45 ± 0.11 vs. 0.89 ± 0.05, ** *p* < 0.01). As the round of sorting increases, the ratio of ITGA6-positive cells also increases (post-first sorting vs. post-multiple sorting: 0.68 ± 0.02 vs. 0.89 ± 0.05, * *p* < 0.05). *p* values were analyzed by one-way ANOVA.

**Figure 4 bioengineering-10-00410-f004:**
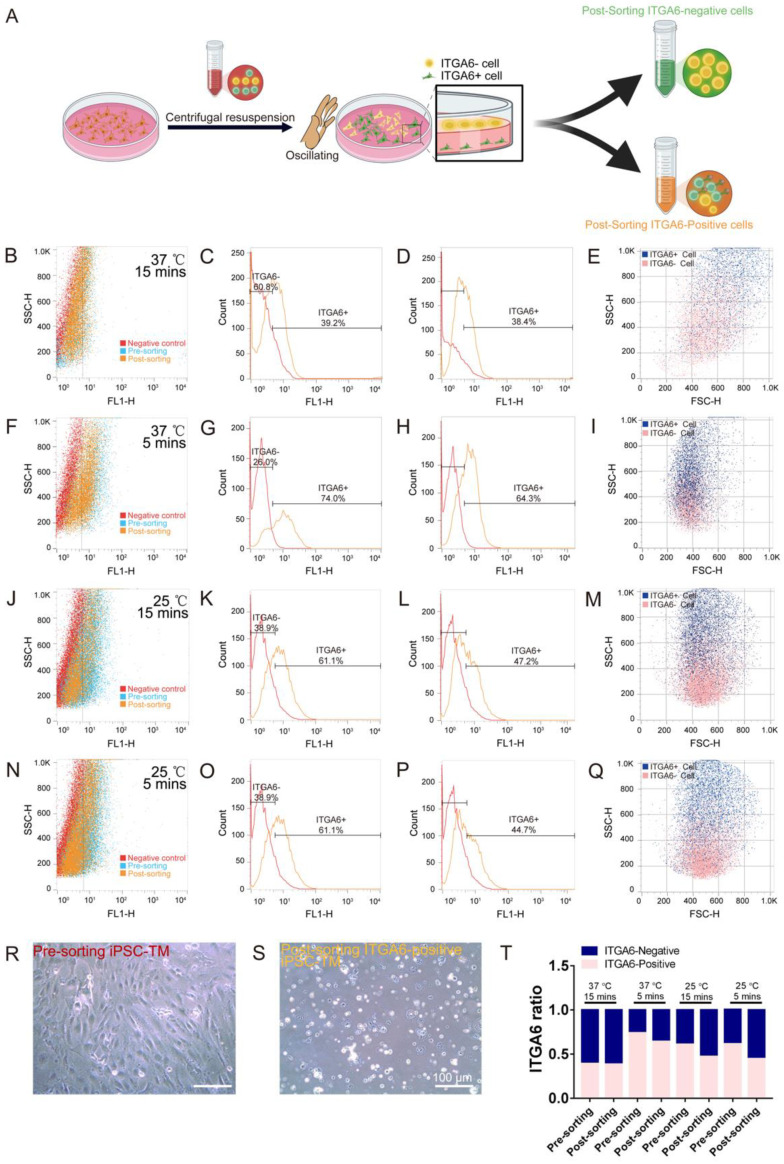
The IP-based purification of ITGA6-positive iPSC-TM. (**A**) Schematic illustration of the ITGA6-positive purification process by IP. iPSC-TM was seeded on a dish pre-coated with the ITGA6 antibody. After oscillation, ITGA6-positive iPSC-TM cells (green) adhered to the dish, while the ITGA6-negative iPSC-TM (yellow) cells floated in the medium. (**B**–**Q**) The ratio of ITGA6-positive cells after IP-based sorting by oscillation at different conditions: (**B**–**E**) 37 °C every 15 min; (**F**–**I**) 37 °C every 5 min; (**N**–**Q**) 25 °C every 15 min; or (**J**–**M**) 25 °C every 5 min. (**R**,**S**) Representative images of the cell morphology of pre-sorted iPSC-TM and ITGA6-positive iPSC-TM after IP. Scale bars, 100 µm. (**T**). Quantification analysis of the ratios of the ITGA6-positive cells (pink) and ITGA6-negative cells (blue).

**Figure 5 bioengineering-10-00410-f005:**
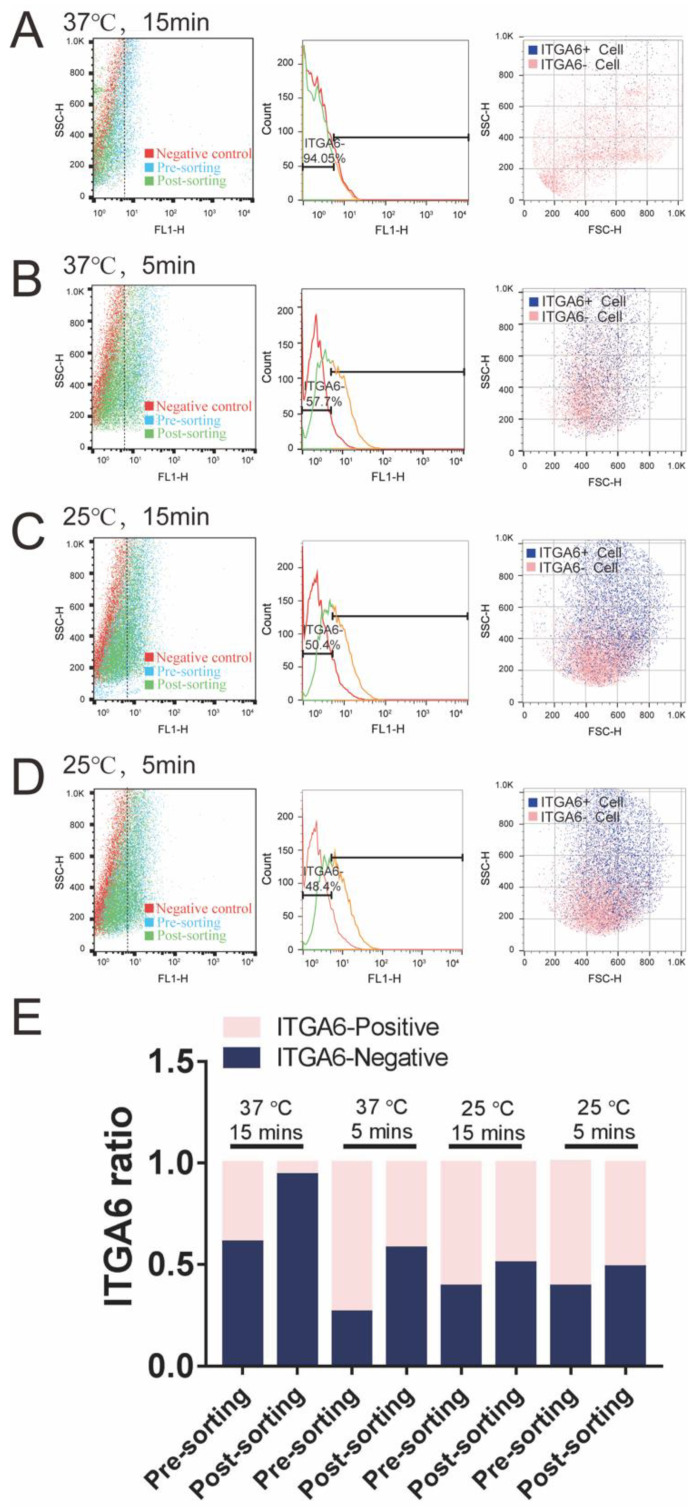
The IP-based purification of ITGA6-negative iPSC-TM. The ratio of the ITGA6-negative cells after the IP-based sorting by oscillation at different conditions: (**A**) at 37 °C every 15 min; (**B**) at 37 °C every 5 min; (**C**) at 25 °C every 15 min; and (**D**) at 25 °C every 5 min. (**E**). Quantification analysis of the ratios of the ITGA6-positive cells (pink) and ITGA6-negative cells (blue).

**Table 1 bioengineering-10-00410-t001:** Human donor demographic characteristics. N/A = not available.

Donor	Age	Gender	Race	Cause of Death
Donor5	80	Male	Caucasian	Acute respiratory distress
Donor6	37	Female	Caucasian	Acute liver failure
Donor8	N/A	N/A	Chinese	N/A

## Data Availability

Not applicable.

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
