# Peer review of "Application of a Magnetic Platform in α6 Integrin-Positive iPSC-TM Purification"

_bioengineering, 2023, doi:10.3390/bioengineering10040410_

Round 1

Reviewer 1 Report (Previous Reviewer 1)

The authors have addressed all of my concerns.

Reviewer 2 Report (Previous Reviewer 2)

Great job with the revisions.  Very solid contribution.  

Reviewer 3 Report (Previous Reviewer 3)

If no other concerns are raised by other reviewers,  I think we can say "accept".

This manuscript is a resubmission of an earlier submission. The following is a list of the peer review reports and author responses from that submission.

Round 1

Reviewer 1 Report

Feng et al present a a protocol based on a magnetic-activated cell sorting (MACS) system and an immunopanning (IP) method for sorting integrin sub-unit alpha 6 (ITGA6)-positive iPSC-TM. This work is important because of the heterogeneity of iPSC-TM cells which impedes the understanding of how the decellularized TM may be regenerated. Overall, the manuscript is well written and supported by the data. However, the authors should include an explanation of why they chose ITGA6 as the cell surface marker used to sort? What is the significance of this subpopulation and is this subpopulation truly just one cell type? Will other cell surface markers would efficiently separate out subpopulations?

Author Response

Response: Thanks for your important comments and careful review.

It is very important of your questions: i) why did we choose ITGA6 as our selection biomarker? ii) what’s the function of this subpopulation of iPSC-TM? iii) how many cell types are in this ITGA6+ iPSC-TM cluster? All these questions have been addressed in our subsequent study, which will be submitted soon. Shortly, we demonstrated that iPSC-TM could efficiently restore aqueous humor outflow and maintain intraocular pressure homeostasis. But, primary TM cells don’t have this function. So, we conducted single cell RNA sequencing to characterize the molecular mechanisms underlying this phenomenon and found an ITGA6+ iPSC-TM cluster. Although we don’t know how many cell types are in this cluster, our pathway enrichment analysis showed that this cluster is enriched with pathways related to integrin cell surface interactions, extracellular matrix organization, and transcriptional regulation of pluripotent stem cells.

Regarding “will other cell surface markers would efficiently separate out subpopuation?”, the short answer is “yes”. This is also our goal to purify different iPSC-TM subpopulations by using different cell surface markers. We have added this advantage of the MACS-based system to the revised manuscript (lines 305-309). Meanwhile, care should be taken to ensure the robust expressions of the selected markers and the proper incubation duration of antibodies to avoid receptor-mediated internalization. It has also been added to the revised manuscript (lines 359-363).

Thank you so much for your important questions.

Revisions to the manuscript (labeling in red):

Line 305: “Our data using the MACS-based approach to purify ITGA6-positive iPSC-TM suggested that this purification method may be feasible to isolate the above subpopulations using different cell surface markers and investigate the roles of different cell clusters in controlling AH outflow and regulating IOP homeostasis.”

Line 359: “The third limitation of these antibody purification methods is receptor-mediated antibody internalization[52], which would impair their effectiveness in cell purification. Our approach should be optimized based on the biological half-life of α6 integrin. Alternatively, anti-β1 integrin antibodies could be involved to stabilize the heterodimer on the cell surface[53].”

Reviewer 2 Report

Feng and coworkers show in this interesting manuscript that magnetic-17 activated cell sorting (MACS) system or an immunopanning (IP) methods can be used for sorting a6-positive iPSC-TM subpopulations.

The manuscript is moderately well-written, supported by the data and would be of general interest to the bioengineering community.  The ability to retrieve stem cells by surface expression of this receptor would have a broad impact since it resides on up to 30 types of stem cells.  The areas of improvement are the following:

1.     The modest change in the recovery of live a6 integrin expressing cells between the two techniques perhaps is not surprising since engagement of the receptor with antibody will result in internalization of the receptor.   The biological half-life of the receptor is approximately 30 minutes---so internalization would affect recovery given the experimental design.  Has this factor been considered in the approach? There should be some comment about this factor as it limits the utility of the approach. 

2.     The use of primaquine as a reversible and non-lethal way to block internalization may be a possible way to improve the recovery.  Alternatively, b1 integrin specific antibodies that stabilize the heterodimer on the surface may be used.

3.     The use of trypsin to retrieve the a6 expressing cells is a limitation of the study since this results in the removal of the N-terminal domain (ligand binding region) of the a6 integrin on live cells.

4.     The writing of the discussion needs to be improved as many of the sentences are awkwardly written and difficult to follow.  Proof reading for grammar changes would improve it. 

The title uses ITGA6 which is the gene name---not the protein name.  Since the manuscript investigates the protein and not the gene---the title should either use CD49f or a6 integrin.

Author Response

Reviewer 2

Feng and coworkers show in this interesting manuscript that magnetic-17 activated cell sorting (MACS) system or an immunopanning (IP) methods can be used for sorting a6-positive iPSC-TM subpopulations. The manuscript is moderately well-written, supported by the data and would be of general interest to the bioengineering community. The ability to retrieve stem cells by surface expression of this receptor would have a broad impact since it resides on up to 30 types of stem cells. The areas of improvement are the following:

  1. The modest change in the recovery of live a6 integrin expressing cells between the two techniques perhaps is not surprising since engagement of the receptor with antibody will result in internalization of the receptor. The biological half-life of the receptor is approximately 30 minutes---so internalization would affect recovery given the experimental design. Has this factor been considered in the approach? There should be some comment about this factor as it limits the utility of the approach.
  2. The use of primaquine as a reversible and non-lethal way to block internalization may be a possible way to improve the recovery. Alternatively, b1 integrin specific antibodies that stabilize the heterodimer on the surface may be used.

Response: It is a great point! Thank you for your comments and suggestions. This limitation has been discussed in the revised manuscript (lines 359-363).

Revisions to the manuscript (labeling in red):

Line 359: “The third limitation of these antibody purification methods is receptor-mediated antibody internalization[52], which would impair their effectiveness in cell purification. Our approach should be optimized based on the biological half-life of α6 integrin. Alternatively, anti-β1 integrin antibodies could be involved to stabilize the heterodimer on the cell surface[53].”

  1. The use of trypsin to retrieve the a6 expressing cells is a limitation of the study since this results in the removal of the N-terminal domain (ligand binding region) of the a6 integrin on live cells.

Response: Thanks. We have revised the limitations in the Discussion (lines 363-367).

Revisions to the manuscript:

Lines 363: “Another factor that may influence the effectiveness of these methods was the utilization of trypsin when we collected cells. As reported, trypsin can lead to the removal of the N-terminal domain of α6 integrin and thus may affect the antibody binding sites[54]. Enzyme-free release of iPSC-TM should be considered.”

  1. The writing of the discussion needs to be improved as many of the sentences are awkwardly written and difficult to follow. Proof reading for grammar changes would improve it.

Response: The current version of this manuscript has undergone substantial revisions (lines 295-345) that hopefully address this concern.

Revisions to the manuscript:

Line 295:

“Based on these findings, this MACS-based purification could be applied as follows.

  1. In the early 1950s, the TM was discovered as an elaborate and complex tissue anatomically[31]. In short, it comprises three distinct layers with different structures and functions in aqueous humor drainage, indicating the heterogeneity of the TM. Recently, two new studies using scRNA-seq verified this heterogeneity, demonstrating 12 types of cells in the conventional outflow tissue[17, 18]. Although researchers have investigated different roles of cells in juxtacanalicular connective tissue or uveal meshwork[32, 33], the functions of many other cell types, such as Schwann cells, melanocytes, and T cells, are still largely unknown. Our data using the MACS-based approach to purify ITGA6-positive iPSC-TM suggested that this purification method may be feasible to isolate the above subpopulations using different cell surface markers and investigate the roles of different cell clusters in controlling AH outflow and regulating IOP homeostasis.
  2. Loss of TM cellularity, aberrant extracellular matrix remodeling, changes in the biomechanical properties of the TM, and mutations have been reported as risk factors for glaucoma [34, 35]. For example, the first pathogenic mutation for primary open-angle glaucoma has been identified in the myocilin gene [36]. The aggregation of mutant myocilin can lead to a severe decline in TM cellularity due to endoplasmic reticulum stress [37, 38]. To this end, it is very important to investigate how damage occurs in different TM subpopulations. Our MACS-based purification provides a simple method to generate TM subpopulations of glaucoma, which may facilitate us in determining the dysfunctional TM subpopulation at the earliest stage. The study can benefit not only the diagnoses but also the treatments for glaucoma patients at the early stages.
  • In recent times, some new glaucoma drugs have been identified, which function primarily by modulating the TM cytoskeleton and the contractile tone of TM cells, their volume, and extracellular matrix deposition, such as Rho kinase inhibitors[39], nitric oxide (NO) signaling regulators[40], latrunculins[41], and ion channel regulators [42]. Besides the pharmacologic treatments, gene therapy also holds a great promise to rescue TM dysfunction. [43-46]. However, which subpopulations of the TM could be efficiently regulated by these new treatments are still elusive. The other side of answering this question would benefit the discovery of the proper delivery approaches for these new drugs/gene therapies. To this end, our MACS-based approach is feasible to address this question.
  1. Moreover, we applied iPSC-TM in regenerating the damaged TM of several glaucoma models, including Tg-MYOCY437H mice[13, 14], GCα1-/- mice, and aged human eyes[15]. As investigated previously, a common phenomenon of iPSC-TM after cell transplantation is that endogenous TM cells could be stimulated to proliferate. Besides iPSCs, mesenchymal stem cells (MSCs) are also used in TM regeneration [47, 48]. MSCs exist in the TM and are identified by analyzing the expressions of stem cell biomarkers [49]. In glaucoma animal models, the transplanted MSCs exhibit positive therapeutic effects on TM regeneration [47, 50, 51], including migration into the TM, secretion factors to recruit nesting-positve progenitor, stimulation cell proliferation of endogenous TM cells. Although encouraging, it is still elusive how the transplanted cells stimulate endogenous cell proliferation. Thus, our MACS-based purification could efficiently isolate different iPSC-TM or MSCs subpopulations in which we are interested and facilitate a better understanding of the mechanism of stem cell-based therapy. ”
  2. The title uses ITGA6 which is the gene name---not the protein name. Since the manuscript investigates the protein and not the gene---the title should either use CD49f or a6 integrin.

Response: Thanks. It has been corrected.

Reviewer 3 Report

In this study, the author developed a new experimental protocol (MACS) to purify the iPSC-TM. The method is compared with an IP method and the findings are the MACS protocol can lead to high ratio of ITGA6-positive iPSC-TM and better cell viability than the IP method.  While the induction of a MACS protocol may be critical to the regenerative medicine in the advancement of glaucoma treatment, this manuscript is weakened by the unclear novelty and insufficient data to completely support the conclusions. It is suggested that the authors make these improvements to the manuscript:

1. Unclear novelty:  it is unclear if this study is the first to use MACS method to purify iPSC-TM. If not, please cite and acknowledge other researchers' work; if yes, please clarify if there is any specific step in the methods that distinguish their MACS protocol from others.  Please cite relevant references for other applications using MACS to purify cells. 

2. Please provide necessary references for the induction of iPSC differentiation to TM.  For this pluripotent stem cell (iPSC), is there any reagent/method used to guide iPSC to differentiate into TM specifically?  This question also is related to Fig. 2Q. What is the 'negative control'? Does this result mean that without any sorting, the iPSC culture will lead to very low differentiation rate into TM-like cells?  Please also provide reference(s) and justification why ITGA6 is the only biomarker to confirm TM differentiation.

3. The authors concluded that iPSC-TM after MACS purification has good cell viability. The authors also claimed 'our MSC-based ... and functional TM subpopulations..' (line 333-335). But only qualitative microscopic images were shown. Thus, their results do not fully support this conclusion. More bioassays on the TM function are suggested to support the conclusion as discussed in line 325-333).  It would be great if the authors can demonstrate that the biological/physiological function of iPSC-TM is preserved after the MACS purification.

4. The significance of this study can be further clarified. The authors have discussed some new drugs that have less side effects and high efficiency.  If this is the case, why would iPSC-TM improve the management of the patients? The author should shorten the discussion of section iii (line 345-367, gene therapy) but elaborate more on how iPSC-TM potentially could help understand pathology or test new drugs.

5. Please add a section of 'Statistical analysis' in Methods to provide more details about statistics. 

6. It is unclear why the authors did not present a similar figure as in Fig. 3F to show ITGA6 ratio quantified from IP method. It would be convincing to show the different purification ratios generated by MACS and IP methods to better support the conclusion.

7. The authors have introduced several recent cell separation methods (line 57-60). The advantage and justification of MACS method was described, but it is unclear why IP separation was chosen as the comparison in this study.   Please provide a brief justification on this aspect. 

Author Response

Reviewer 3

In this study, the author developed a new experimental protocol (MACS) to purify the iPSC-TM. The method is compared with an IP method and the findings are the MACS protocol can lead to high ratio of ITGA6-positive iPSC-TM and better cell viability than the IP method. While the induction of a MACS protocol may be critical to the regenerative medicine in the advancement of glaucoma treatment, this manuscript is weakened by the unclear novelty and insufficient data to completely support the conclusions. It is suggested that the authors make these improvements to the manuscript:

  1. Unclear novelty: it is unclear if this study is the first to use MACS method to purify iPSC-TM. If not, please cite and acknowledge other researchers' work; if yes, please clarify if there is any specific step in the methods that distinguish their MACS protocol from others. Please cite relevant references for other applications using MACS to purify cells.

Response: Thanks for your suggestions. The MACS-based approach has been applied in our previous studies to deplete SSEA-1-positive iPSC-TM cells out of the other iPSC-TM cells. As suggested, these studies have been cited. The negative selection is achieved using LD columns (Miltenyi Biotec, 130-048-901; Void volume: 1.35 mL, Reservoir volume: 7 mL) developed for the depletion of cells with a low level of marker expression. Compared to LD columns, LS columns (Miltenyi Biotec, 130-048-401; Void volume: 400 μL, Reservoir volume: 8 mL) generate lower affinity to the magnetic cells, which can be used for the positive selection of cells that strongly express the magnetically labeled surface antigen. We applied this technique in ITGA6-positive iPSC-TM purification in this study and tested the purification efficiency. To our knowledge, this is the first time applying LS columns in the positive selection of iPSC-TM cells. Based on your suggestions, the novelty of this study has been clarified in the revised manuscript (lines 142-147; 271-279).

Revisions to the manuscript (labeling in red):

Line 142: “The labeled cells on the microbeads were purified by passing through the LS column (Miltenyi Biotec, 130-048-401; Void volume: 400 μL, Reservoir volume: 8 mL) or LD column (Miltenyi Biotec, 130-048-901; Void volume: 1.35 mL, Reservoir volume: 7 mL), which was pre-placed on a magnetic separation rack (Miltenyi Biotec, 130-042-501) and pre-washed according to the manufacturer’s instructions, and named ITGA6-positive/negative iPSC-TM.”

Line 271: “MACS-based purification has also been widely used in sorting many cell types due to its simplicity, cheapness, and convenience. We have previously demonstrated a viable immunomagnetic sorting approach to depleting the undifferentiated cells from iPSC-TM based on the robust expression of SSEA-1 by stem cells using the LD columns[13, 14, 16]. Compared to the LD columns, the LS columns can be used for the positive selection of cells that strongly express the magnetically labeled surface antigen. We employed this technique (Fig. 2: positive selection using the LS columns; Fig. 3: negative selection using the LD columns) to purify ITGA6-positive/negative iPSC-TM cells.”

  1. Please provide necessary references for the induction of iPSC differentiation to TM. For this pluripotent stem cell (iPSC), is there any reagent/method used to guide iPSC to differentiate into TM specifically? This question also is related to Fig. 2Q. What is the 'negative control'? Does this result mean that without any sorting, the iPSC culture will lead to very low differentiation rate into TM-like cells? Please also provide reference(s) and justification why ITGA6 is the only biomarker to confirm TM differentiation.

Response: Thanks. Based on your suggestions, the references have been added. We applied the medium conditioned by primary TM cells (at least three donors) for differentiating iPSCs towards TM-like cells.

The negative control in Fig. 2Q is the differentiated iPSCs, named iPSC-TM in this study, without any sorting. ITGA6 is not the biomarker to confirm TM differentiation. Instead, it is a biomarker for a specific iPSC-TM cluster. Previously, we have demonstrated that iPSC-TM can efficiently restore aqueous humor outflow and maintain intraocular pressure homeostasis. But, primary TM cells don’t have this function. So, we conducted single cell RNA sequencing to characterize the molecular mechanisms underlying this phenomenon and found an ITGA6+ iPSC-TM cluster. In this study, we aimed to use ITGA6 as an example to test whether the MACS-based system or IP method could be applied to purify the specific iPSC-TM cells out of the heterogenous iPSC-TM mixture. Thank you so much for your comments. All these questions have been addressed in our subsequent study, which will be submitted soon. We also hope to link this paper to our subsequent study, which would benefit our readers to better understand the importance of this technique.

Revisions to the manuscript:

Line 116: “Medium conditioned by hTM of donors 13 were collected, sterilized using a mixed cellulose ester membrane with 0.2 µm pores (Millipore, MA, USA), and used for human iPSCs differentiation[10, 13, 14, 16].”

  1. The authors concluded that iPSC-TM after MACS purification has good cell viability. The authors also claimed 'our MSC-based ... and functional TM subpopulations..' (line 333-335). But only qualitative microscopic images were shown. Thus, their results do not fully support this conclusion. More bioassays on the TM function are suggested to support the conclusion as discussed in line 325-333). It would be great if the authors can demonstrate that the biological/physiological function of iPSC-TM is preserved after the MACS purification.

Response: It is a great point. As suggested, we employed Trypan Blue Staining Assay and analyzed cell viability after the MACS purification by Countstar (Alit Biotech Co., Ltd., Shanghai, China). This result has been added to the revised manuscript (lines 151-153; 282-285).

Revisions to the manuscript:

Line 151: “The purified ITGA6-positive iPSC-TM cells were incubated with 0.4% (wt/vol) filtered trypan blue solution at room temperature for 3 minutes. Cell viability was analyzed using Countstar (Alit Biotech Co., Ltd., Shanghai, China).”

Line 282: “In comparison with multiple rounds of purification, cells are still in a relatively healthy condition after one round of purification, like pre-sorted iPSC-TM (Fig. 2; cell viability: 91.7% by Trypan Blue Staining Assay).”

  1. The significance of this study can be further clarified. The authors have discussed some new drugs that have less side effects and high efficiency. If this is the case, why would iPSC-TM improve the management of the patients? The author should shorten the discussion of section iii (line 345-367, gene therapy) but elaborate more on how iPSC-TM potentially could help understand pathology or test new drugs.

Response: Compared to the traditional drugs, current pharmacologic treatments targeting the conventional outflow pathway do regulate IOP homeostasis more efficiently with less side effects. However, loss of TM cells has become a common phenomenon in glaucoma, and these treatments can only target the retained TM. To this end, it is of particular interest to test whether cell-based therapy to restore TM cellularity in glaucoma may be an alternative approach to treat glaucoma. Another advantage of cell-based therapy is that we might treat glaucoma patients by one-time cell transplantation instead of the daily use of glaucoma drugs.

Based on your suggestion, we have shortened the discussion of section iii.

Revisions to the manuscript:

Line 321: “In recent times, some new glaucoma drugs have been identified, which function primarily by modulating the TM cytoskeleton and the contractile tone of TM cells, their volume, and extracellular matrix deposition, such as Rho kinase inhibitors[39], nitric oxide (NO) signaling regulators[40], latrunculins[41], and ion channel regulators [42]. Besides the pharmacologic treatments, gene therapy also holds a great promise to rescue TM dysfunction. [43-46]. However, which subpopulations of the TM could be efficiently regulated by these new treatments are still elusive. The other side of answering this question would benefit the discovery of the proper delivery approaches for these new drugs/gene therapies. To this end, our MACS-based approach is feasible to address this question.”

  1. Please add a section of 'Statistical analysis' in Methods to provide more details about statistics.

Response: Thanks for your comments. It has been added to the revised manuscript (lines 175-177).

Revisions to the manuscript:

Line 175: “One-way ANOVA was performed for statistical analysis of ITGA6-positive/negative ratios after positive/negative selection. Data were expressed as the mean ± SD. p values < 0.05 were considered to be statistically significant.”

  1. It is unclear why the authors did not present a similar figure as in Fig. 3F to show ITGA6 ratio quantified from IP method. It would be convincing to show the different purification ratios generated by MACS and IP methods to better support the conclusion.

Response: Yes. It has been added to Fig. 4 and 5.

Revisions to the manuscript:

Line 623:

Figure 4.

Line 637:

Figure 5.

  1. The authors have introduced several recent cell separation methods (line 57-60). The advantage and justification of MACS method was described, but it is unclear why IP separation was chosen as the comparison in this study. Please provide a brief justification on this aspect.

Response: Thanks for your comments. It has been added to the revised manuscript (lines 76-80).

Revisions to the manuscript:

Line 76: “The IP technique is an alternative immuno-approach that purifies cells using cell culture plates that have been pre-coated with some specific antibodies (anti-cell surface proteins). This approach could yield retinal ganglion cells at higher viability than the magnetic sorting technique[21, 25].”
